# Reintervention for Acute Aortic Prosthesis Endocarditis: Early and Mid-Term Outcomes

**DOI:** 10.3390/jcm13247690

**Published:** 2024-12-17

**Authors:** Michele D’Alonzo, Yuthiline Chabry, Giovanna Melica, Sébastien Gallien, Pascal Lim, David Aouate, Raphaëlle Huguet, Adrien Galy, Raphaël Lepeule, Vincent Fihman, Claire Pressiat, Thierry Folliguet, Antonio Fiore

**Affiliations:** 1Service de Chirurgie Cardiaque, DMU CARE, Assistance Publique-Hôpitaux de Paris (AP-HP), Hôpitaux Universitaires Henri Mondor, F-94010 Créteil, France; yuthiline@gmail.com (Y.C.); thierry.folliguet@aphp.fr (T.F.); antonio.fiore@aphp.fr (A.F.); 2Faculté de Santé, Université Paris Est Créteil, F-94010 Créteil, France; 3Service de Maladies Infectieuses et Immunologie Clinique, Assistance Publique-Hôpitaux de Paris (AP-HP), Hôpitaux Universitaires Henri Mondor, Faculté de Santé, Université Paris Est Créteil, F-94010 Créteil, France; giovanna.melica@aphp.fr (G.M.); sebastien.gallien@aphp.fr (S.G.); 4Service de Cardiologie, DMU Médecine, Assistance Publique-Hôpitaux de Paris (AP-HP), Hôpitaux Universitaires Henri Mondor, Faculté de Santé, Université Paris Est Créteil, F-94010 Créteil, France; pascal.lim@aphp.fr (P.L.); davidrobin.aouate@aphp.fr (D.A.); raphaelle.huguet@aphp.fr (R.H.); 5Unité Transversale de Traitement des Infections, DMU PDTI, Assistance Publique-Hôpitaux de Paris (AP-HP), Hôpitaux Universitaires Henri Mondor, Faculté de Santé, Université Paris Est Créteil, F-94010 Créteil, France; adrien.galy@aphp.fr (A.G.); raphael.lepeule@aphp.fr (R.L.); 6Unité de Bactériologie, DMU PDTI, Assistance Publique-Hôpitaux de Paris (AP-HP), Hôpitaux Universitaires Henri Mondor, Faculté de Santé, Université Paris Est Créteil, F-94010 Créteil, France; vincent.fihman@aphp.fr; 7Université Paris Est, EA 7380 Dynamyc, Ecole Nationale Vétérinaire d’Alfort, USC Anses, F-94010 Créteil, France; 8Laboratoire de Pharmacologie, DMU Biologie-Pathologie, Assistance Publique-Hôpitaux de Paris (AP-HP), Hôpitaux Universitaires Henri Mondor, Faculté de Santé, Université Paris Est Créteil, F-94010 Créteil, France; claire.pressiat@aphp.fr; 9Inserm U955-IMRB, Equipe 03 “Pharmacologie et Technologies pour les Maladies Cardiovasculaires (PROTECT)”, Ecole Nationale Vétérinaire d’Alfort (EnVA), Université Paris Est Créteil, F-94700 Maisons-Alfort, France; 10Université Paris Est Créteil, Inserm, IMRB U955, CEpiA Team, F-94000 Créteil, France

**Keywords:** aortic valve, endocarditis, redo surgery, cardiac surgery challenges

## Abstract

**Objective:** This study aimed to analyze the outcomes and challenges associated with surgical redo procedures following aortic valve replacement for acute infective endocarditis. While transcatheter aortic valve implantation is growing in terms of its utilization for degenerative bioprostheses failure, valve-in-valve procedures are limited in acute aortic endocarditis. Surgical interventions for aortic prosthesis endocarditis carry a significant risk, with a higher mortality and morbidity, often requiring concomitant complex procedures. **Methods:** This was a retrospective, monocentric, observational study. We identified 352 patients with infective endocarditis from the institutional database. After applying the inclusion and exclusion criteria, 54 patients who underwent surgical re-operation between 2016 and 2023 were included. Endpoints included early and late mortality, complications, and major adverse cardiac and cerebrovascular events (MACCEs). **Results:** From the cohort, predominantly male and with an average age of 71.9 ± 12.1 years old (79.6%), the following notable findings were derived: isolated aortic valve replacement was feasible only in 34 patients (63%) while more complex procedures were demanded in the other cases; the overall 30-day mortality rate was 18.5%, post-operative ECMO occurred in 9.3% of cases, and post-operative new stroke in 2.7%; the 5-year overall survival rate was 58.3 ± 18.6%, while freedom from MACCEs was 41.7 ± 19.7%. Another re-intervention was required in three patients during follow-up, with one case attributed to re-endocarditis. **Conclusions:** Despite advancements in surgical and perioperative care, redo procedures for acute infective endocarditis pose significant risks, as evidenced by the high 30-day mortality rate. However, the 5-year survival suggests a relatively acceptable outcome, underscoring the complexities and challenges inherent in managing this condition surgically.

## 1. Introduction

Acute endocarditis affecting aortic valve prostheses presents a formidable challenge in contemporary cardiology, underscored by significant morbidity and mortality rates. The comprehensive management of this condition requires the meticulous consideration of treatment strategies, prognostic factors, and microbiological etiology to optimize patient outcomes. The incidence of prosthetic valve endocarditis (PVE) remains elusive, constituting approximately 10% to 30% of all cases of infective endocarditis and affecting 1% to 6% of patients with valve prostheses, regardless of valve type [1,2,3].

Early research has shed light on the substantial impact of PVE on early mortality rates, with independent predictors including the presence of PVE itself and the severity of heart failure at presentation. Moreover, a paradigm shift has been proposed, suggesting potential benefits in delaying surgical intervention to allow for a healed state in PVE cases, except when urgent intervention is warranted [4].

Treatment modalities, encompassing medical and surgical approaches, play a pivotal role in managing PVE. While antibiotic therapy remains fundamental, surgical intervention has demonstrated superior long-term survival outcomes, advocating for its prudent use in the appropriate cases [5]. The microbiological landscape of PVE further complicates treatment decisions, with distinct patterns observed between mechanical and bioprosthetic valves, particularly concerning the prevalence of Staphylococcus aureus infections [6]. Staphylococcus aureus infection, in particular, is associated with a poorer prognosis, necessitating meticulous management. While uncomplicated cases may benefit from conservative management, prompt surgical intervention is typically indispensable in more complex scenarios to mitigate adverse outcomes [7].

Traditionally, PVE has been associated with higher mortality rates compared to native aortic valve endocarditis [7,8], especially in the presence of complications such as heart failure, abscess formation, and stroke. However, recent studies have challenged this concept, suggesting comparable outcomes between patients undergoing cardiac surgery for native aortic valve endocarditis and those with aortic PVE [9,10].

Contemporary research underscores the evolving clinical profile of PVE and the related complications, including heart failure and persistent bacteremia, underscoring the necessity of a multidisciplinary approach to optimize patient care [11].

The aim of this study was to examine outcomes related to our experience concerning PVE surgical management. This assessment encompasses both immediate postoperative results and mid-term occurrences.

## 2. Materials and Methods

We identified 352 patients with infective endocarditis from the institutional database. For this study, patients consecutively undergoing redo cardiac surgery for acute endocarditis involving the aortic prosthesis were enrolled. Exclusion criteria were age < 18 years old and non-aortic prosthesis endocarditis (Figure 1).

Data were retrospectively collected from January 2016 to December 2023 in “Henri Mondor Hospital”, Créteil, France, which is a part of Assistance Publique-Hopitaux de Paris (APHP). Specifically, this hospital specializes in the management of endocarditis, with a closely monitored diagnostic procedure and with the follow-up process involving weekly multidisciplinary sessions as suggested by the 2015 ESC guidelines [12]. These sessions engage not only cardiothoracic surgeons but also infectious disease specialists, cardiologists, gastroenterologists, and other relevant specialists.

Diagnosis of PVE was based on clinical findings (fever, inflammatory syndromes), laboratory testing (blood cultures, culture or 16S rRNA amplicon-sequencing performed on vegetation or cardiac tissue, Coxiella burnetii or Bartonella species serologies, leukocytosis, and levels of C-reactive protein and procalcitonin), results of transthoracic/transesophageal echocardiography, and intraoperative findings.

All operations were performed through a median full sternotomy, standard cardiopulmonary bypass (CPB), and cold blood or crystalloid cardioplegia.

All data, including demographic characteristics, primary hospitalization outcomes, and post-discharge events, were meticulously gathered from an institutional database. Data were sourced from patients’ medical records or obtained through telephone interviews. The follow-up time was calculated either to death or to the last verified contact with the patient. Follow-up data were collected until February 2024.

The primary endpoints included early and late mortality, while secondary endpoint involved a composite analysis, encompassing the incidence of complications and major adverse cardiac and cerebrovascular events (MACCEs was defined as death, disabling stroke, re-endocarditis, permanent pacemaker implantation, and myocardial infarction).

For the latter outcome, the fourth universal definition of myocardial infarction was used [13].


**Statistical analysis**


Distribution normality of continuous variables was tested with Kolmogorov–Smirnov test. Data conforming to normal distribution were expressed as the mean ± standard deviation (SD), while non-normal distribution was described as the median [interquartile range]. Student’s t-test and Kruskal–Wallis tests were used for normal and non-normal intergroup comparisons, respectively. Categorical variables were expressed as frequency (percentage) and compared with the Pearson Chi-square test or Fisher’s exact test when the expected frequencies of one or more cells were less than 5.

The estimated survival probability was analyzed with the Kaplan–Meier method. A *p*-value ≤ 0.05 was considered statistically significant. Microsoft Office Excel (Microsoft, Redmond, WA, USA) was used for data extraction, and all analyses were performed in R, version 4.3.1 (R Software for Statistical Computing, Vienna, Austria) within RStudio.

## 3. Results

During the specified period, 352 patients underwent cardiac surgery after diagnosis of endocarditis. Among them, 180 patients manifested aortic valve involvement, with 126 cases affecting the native valve and 54 cases involving a previously implanted aortic prosthesis (Figure 1).
Figure 1Flowchart showing the patient recruitment and exclusion process.
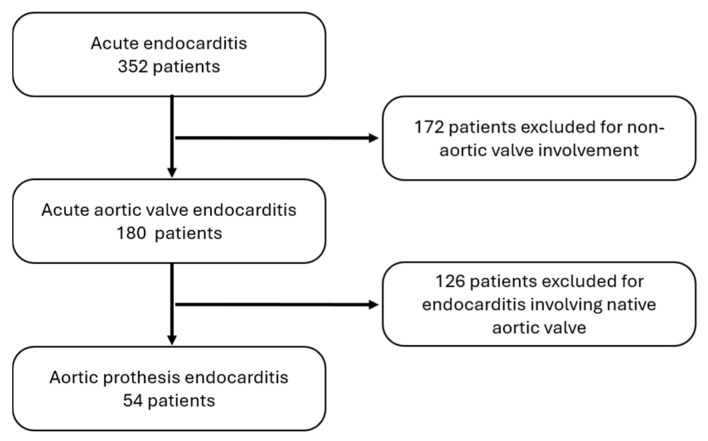


A comprehensive analysis was conducted on a cohort of 54 patients who underwent redo cardiac surgery due to acute aortic prosthesis endocarditis, constituting the focus of this study. The demographic and clinical characteristics of patients are succinctly summarized in Table 1. Notably, a predominant proportion of the enrolled patients were male, comprising 79.6% of the study population, while systemic arterial hypertension emerged as the prevailing comorbidity. The average age of the cohort stood at 71.9 ± 12.1 years, accompanied by a EuroSCORE II of 15.0 ± 10.9%, reflecting the collective burden of surgical risk among the participants. Preoperative echocardiography revealed the presence of large vegetations (>10 mm) in half of the patients (29 patients, 53.7%), annular abscess (31 patients, 57.4%), and aortic fistula (2 patients); meanwhile, 8 patients (14.8%) presented abnormalities in cardiac rhythm conduction involving the atrioventricular (AV) bundle. Additional data regarding the inflammatory and endocrinological status of the patients are presented in Table 1.


**Operative and early outcomes**


Table 2 depicts the operative data and early outcomes. The vast majority of patients underwent urgent cardiac surgery (47 out of 54 patients, 87.0%). The most common surgical procedure was isolated aortic valve replacement with or without abscess exclusion, feasible in 34 patients (63%). The Bentall procedure was carried out in 12 patients (22%) while concomitant mitral surgery was necessary in 6 patients (11%). The most extensive procedure involving aortic valve or root replacement and mitral valve replacement, along with the reconstruction of the aortomitral fibrous body, i.e., the Commando procedure, was performed in two patients (4%). The mean aortic cross-clamping time was 150.5 ± 68.5 min, while the mean cardiopulmonary bypass time was 207.7 ± 109.6 min.

Mechanical circulatory support (ECMO) was required postoperatively or within 48 h in five patients (9.3%). Prolonged inotropic support (>12 h) was necessary for 23 patients (42.6%), and nearly all patients received at least one transfusion (40 patients, 74.1%). Mechanical ventilation support was required for 10 patients (18.5%), and 22 patients (40.7%) had an intensive care unit (ICU) stay exceeding 2 days. During the main hospitalization, adverse events were relatively infrequent; stroke occurred in 2 patients (3.7%), pacemaker implantation in 12 patients (22.2%), and temporary renal replacement therapy was utilized in 8 patients (14.8%). The 30-day mortality rate was 18.5% (10 patients). Recorded causes of death include cardiogenic shock followed by multi-organ failure in half of the deceased patients, with acute liver failure, septic shock, and massive bleeding registered in the other five patients.


**Mid-terms outcomes**


The mean follow-up was 741 days. The 1-, 3- and 5-year overall survival were 77.3 ± 11.4%, 65.5 ± 14.5%, and 58.3 ± 18.6%, respectively (Figure 2).

The 1-, 3- and 5- year the freedom from MACCEs were 75.1 ± 11.8%, 58.0 ± 15.3% and 41.7 ± 19.7%, respectively (Figure 3). During this observation time, three patients required another surgical reintervention (one for re-endocarditis).

## 4. Discussion

Surgical intervention has greatly enhanced prognosis, initially serving as a follow-up to antibiotic therapy but now also utilized during active disease phases [14].

Septic status, heart failure, cardiac and systemic comorbidities are common features of prosthetic aortic valve endocarditis and contribute to poor patient survival. Surgical treatment is usually recommended in these conditions, but it is still associated with high hospital mortality [15,16,17]. Nevertheless, successful long-term outcomes depend on various factors, such as the effective extension of damage (sometimes different from echocardiography or radiodiagnostic imaging exams), the selection of the surgical technique, procedural details, and the expertise of the healthcare center.

Habib and colleagues showed that heart failure was one of the most prominent factors associated with surgical treatment for prosthetic valve endocarditis, while presentation with cerebral complications was, in more than 10% of cases, the major reason for withholding surgery when indicated [18].

Pericart and colleagues reported that peri-annular abscess and prosthesis dehiscence were more common in patients who underwent valve surgery compared to medically managed patients [15]. In our series, half of the patients presented with a peri-annular abscess, which was one of the most frequent indications for early surgery in the study.

The American Association for Thoracic Surgery (AATS) recommendations follow that when the aortic root and annulus remain intact after radical debridement, implanting a new prosthetic valve is reasonable [19]. In our series, we performed isolated aortic valve replacement with or without peri-annular abscess exclusion in most patients (64%). More aggressive procedures were required in the remaining patients, including the Bentall–De Bono, associated mitral valve surgery, or Commando procedure.

In these challenging surgical scenarios, we registered an acceptable postoperative early mortality rate, stable survival, and low rates of further reoperation and recurrence of infection. Specifically, in our study of 54 patients, we registered 10 deaths (18.5%) within 30 days of the redo surgery. This finding aligns with the results from the International Collaboration on Endocarditis-Prospective Cohort Study that reported an in-hospital mortality of 22% and with the Italian national registry that reported an in-hospital mortality of 19% [20,21].

The causes of death align with those observed in other studies, predominantly featuring heart failure, succeeded by multiorgan failure and septic shock resulting from the inability to manage the systemic infection originating from endocarditis.

In relation to coagulation, it was found that red blood cell transfusions were required in 74.1% of cases, while re-exploration for bleeding was deemed necessary in 20.4% of instances. This notably high occurrence can be attributed to a combination of factors, including the necessity for redo procedures, the intricate and prolonged nature of the surgeries, and the presence of inflammatory conditions which can adversely impact the coagulation cascade.

Concerning definite pacemaker implantation, 12 patients (22.2%) required definite pacemaker implantation. The incidence of this adverse event agreed with other studies like that of Galeone, in which 26 patients out 144 (18%) demonstrated the same need [22].

Regarding the occurrence of other adverse events during hospitalization, no significant differences were noted compared to the current literature regarding native valve endocarditis. Specifically, in terms of neurological complications, among our patient population, 16 out of 54 individuals (30%) had a history of previous cerebrovascular events. Following surgical intervention, all patients underwent postoperative CT scans for monitoring purposes, revealing two new cases of stroke (3.7%), one in a patient with a prior history of stroke and another in a patient with no previous neurological issues. Indeed, it is primarily the neurological aspect rather than the complexity of the surgical procedure or other comorbidities that may contribute to the refusal of a redo procedure despite a surgical indication [18,23,24].

Prosthetic valve endocarditis cannot, in itself, be considered an exclusion criterion for surgery due to its high risk. This is also supported by a recent study by Pizzino, which analyzed 102 patients, half of whom had prosthetic valve endocarditis, and demonstrated no association between the incidence of major adverse events and infective endocarditis on prostheses. Instead, only prosthetic detachment was associated with all-cause mortality [25].

Our mean follow-up was 741 days. The 1-, 3- and 5-year overall survival was 77.3 ± 11.4%, 65.5 ± 14.5%, and 58.3 ± 18.6%, respectively. These results can be interpreted as good in relation to the critical state of patients, the history of disease, the comorbidities, and the complexity of the surgical procedures. Our 5-year survival rate is very similar to that of the study by Weber et al., which performed a propensity score matching between native valve endocarditis and prosthetic valve endocarditis, finding no substantial differences in outcomes at follow-up [26].

However, Perrotta et al. reported on their two-decade experience with surgical treatment of 84 patients with aortic PVE and showed even better long-term survival rates (80% at 5 years and 65% at 10 years), with a significant increase in survival in the second decade of the study [27]. The better long-term survival observed in this latter study could be due to the younger age of the patients (mean age: 58 years) compared to age of patients in our series (mean age: 71.9 years).

Due to the limited population number (54 patients), we were not able to identify any significant independent factor associated with increased operative risk or mid-term mortality.

During our follow-up, the incidence of re-endocarditis was remarkably low, with only one patient experiencing a new episode of endocarditis. To summarize the patient’s medical history, in 2017, the patient underwent elective surgery for aortic valve replacement due to severe valve stenosis, with the implantation of a 23 mm Trifecta bioprosthesis. In July 2020, infectious endocarditis on the bioprosthesis (caused by Streptococcus oralis) led to a 30 × 20 mm peri-prosthetic abscess. Subsequently, the patient underwent a second cardiac surgical procedure involving annulus reconstruction and the implantation of a 23 mm Inspiris Resilia bioprosthesis. In October 2021, the patient was admitted for septic shock, with Staphylococcus haemolyticus and Candida albicans detected. Treatment involved a third cardiac surgical procedure with the implantation of a 21 mm Inspiris Resilia bioprosthesis and the exclusion of the newly formed pseudoaneurysm at the level of the aortic annulus. Unfortunately, the patient passed away during hospitalization for his final cardiac procedure due to multi-organ failure.

The main limitations of our study are its design of a single-center retrospective evaluation and the relatively limited size of our population. Our results were derived from a well-defined population (definite prosthetic aortic valve endocarditis with a surgical indication) that was managed according to a consolidated and shared practice that included an evaluation of mid-term survival. Surgical early mortality is still high, but mid-term survival and freedom from reoperation and recurrence of infection are satisfactory.

## 5. Conclusions

Despite advancements in surgical management and perioperative care, our study confirmed that the operative risk associated with redo procedures following aortic valve replacement for infective endocarditis is high. The 30-day mortality rate underscores the gravity of this condition and the challenges encountered during surgical management; nevertheless, the 5-year survival rate suggests an acceptable outcome.

## Figures and Tables

**Figure 2 jcm-13-07690-f002:**
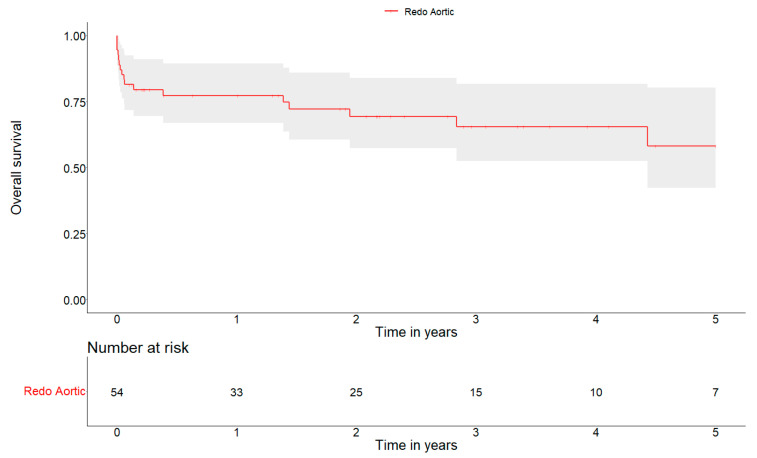
Kaplan–Meier survival curve at 5 years.

**Figure 3 jcm-13-07690-f003:**
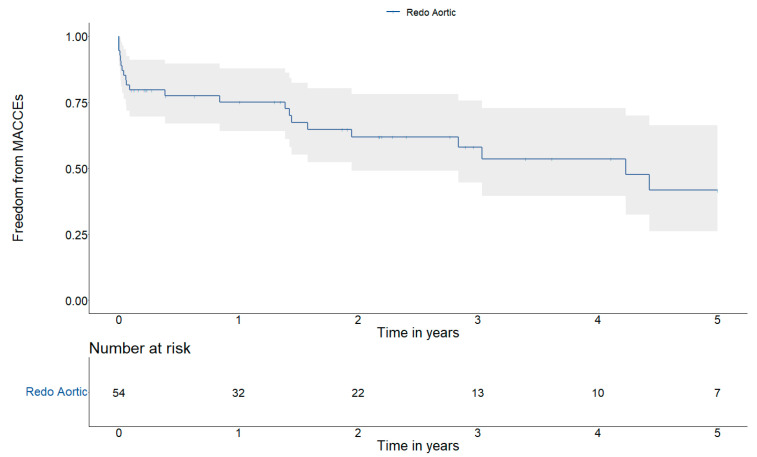
Kaplan–Meier MACCEs curve at 5 years.

**Table 1 jcm-13-07690-t001:** Baseline population characteristics.

**General characteristics**	
Age (years)	71.9 ± 12.1
Gender (male)	43 (79.6)
BMI (kg/m^2^)	26.9 ± 6.9
Systemic hypertension	37 (68.5)
Diabetes mellitus on insulin	6 (11.1)
Glycated hemoglobin (%)	5.9 [0.92]
Glomerular filtration rate (mL/min)	68.3 ± 33.9
EuroSCORE II (%)	15.0 ± 10.9
**Clinical presentation**	
Cardiogenic shock	3 (5.6)
AV conduction disorder	8 (14.81)
Splenic abscess	7 (13.0)
Persistent infection	27 (50)
Cerebral embolism	16 (30)
**Echocardiographic findings**	
Large vegetation (>10 mm)	29 (53.7)
Annular abscess	31 (57.4)
Aortic fistula	2 (3.7)
sPAP (mmHg)	31.1 ± 8.7
LVEF (%)	57.2 ±7.2
**Biological findings**	
Blood hemoglobin (mg/dL)	10.4 ± 1.7
Serum C-reactive protein (ng/dL)	148.5 ± 112.4
Serum pro-calcitonin (ng/dL)	1.4 ± 2.4
**Microbiological findings**	
Staphylococcus aureus	12 (22.2)
Staphylococcaceae (excluded aureus)	5 (9.3)
Enterococcus faecalis	6 (11.1)
Streptococcaceae	18 (33.3)
Others	9 (16.6)
Negative cultures (not identified)	4 (7.5)

Values are expressed in mean ± standard deviation, median [interquartile], or frequency (percentage). BMI: body max index; AV: atrioventricular; sPAP: systolic pulmonary artery pression; LVEF: left ventricular ejection fraction.

**Table 2 jcm-13-07690-t002:** Operative data and early outcomes.

**Intraoperative data**	
Isolated AVR ± abscess exclusion	34 (63%)
Bentall procedure	12 (22%)
Concomitant mitral surgery	6 (11%)
Commando procedure	2 (4%)
CPB (min)	207.7 ± 109.6
ACC (min)	150.5 ± 68.5
**Post-operative data**	
RBC transfusion	40 (74.1)
Re-exploration for bleeding	11 (20.4)
Prolonged use of inotropes (>12 h)	23 (42.6)
ECMO	5 (9.3)
MAV > 48 h	10 (18.5)
ICU stay > 48 h	22 (40.7)
Pacemaker implantation	12 (22.2)
Sternal wound infection	3 (5.6)
Post-operative AF	18 (33.3)
Post-operative stroke	2 (3.7)
Renal replacement treatment	8 (14.8)
**Death within 30-day**	**10 (18.5)**
**Death cause**	
Multi-organ failure	5 (9.3)
Massive bleeding	2 (3.7)
Liver failure	1 (1.9)
Septic shock	2 (3.7)

Values are expressed as mean ± standard deviation or frequency (percentage). AVR: aortic valve replacement; CPB: cardiopulmonary bypass; ACC: aortic cross clamping; RBC: red blood cells; ECMO: extra corporeal membrane oxygenation; MAV: mechanical assisted ventilation; ICU: intensive care unit; AF: atrial fibrillation.

## Data Availability

The raw data supporting the conclusions of this article will be made available by the authors upon request.

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
