# Peer review of "Reintervention for Acute Aortic Prosthesis Endocarditis: Early and Mid-Term Outcomes"

_jcm, 2024, doi:10.3390/jcm13247690_

Round 1
Reviewer 1 Report
Comments and Suggestions for Authors
We would like to express our gratitude to the authors for submitting their manuscript to our journal. The article addresses the critical subject of reintervention for acute aortic prosthesis endocarditis, focusing on early and mid-term outcomes. The results presented are intriguing and conveyed in an optimal manner. However, we have identified several minor revisions that are necessary to enhance the clarity and impact of the manuscript.
Firstly, we recommend the inclusion of a graphical abstract that succinctly summarizes the key findings and messages of the study. This would not only improve the accessibility of the results but also provide readers with a quick overview of the research's significance.
Secondly, we suggest a more comprehensive exploration in the introduction regarding the impact of complications such as heart failure associated with infection on the outcomes of patients undergoing surgical intervention for infective endocarditis. In this context, we recommend citing the work of Pizzino et al. (PMID: 38786960, PMCID: PMC11121817, DOI: 10.3390/jcdd11050138), as it provides valuable insights into this issue.
Furthermore, the abstract should be consolidated to a maximum of 250 words in accordance with the journal's guidelines.
Lastly, if feasible, we advise incorporating a detailed description of the various types of aortic valves implanted at baseline in the sample table, prior to their infection.
Author Response
We would like to express our gratitude to the authors for submitting their manuscript to our journal. The article addresses the critical subject of reintervention for acute aortic prosthesis endocarditis, focusing on early and mid-term outcomes. The results presented are intriguing and conveyed in an optimal manner. However, we have identified several minor revisions that are necessary to enhance the clarity and impact of the manuscript.
Dear Reviewer,
Thank you for taking the time to read and analyze our manuscript. We greatly appreciate your thoughtful comments and positive feedback regarding our study. We also sincerely appreciate your constructive suggestions, which will undoubtedly help enhance the clarity and impact of our manuscript.
Firstly, we recommend the inclusion of a graphical abstract that succinctly summarizes the key findings and messages of the study. This would not only improve the accessibility of the results but also provide readers with a quick overview of the research's significance.
In response to you, we have created a graphical abstract that succinctly summarizes the key findings and messages of our study. We hope it meets your expectations.
Secondly, we suggest a more comprehensive exploration in the introduction regarding the impact of complications such as heart failure associated with infection on the outcomes of patients undergoing surgical intervention for infective endocarditis. In this context, we recommend citing the work of Pizzino et al. (PMID: 38786960, PMCID: PMC11121817, DOI: 10.3390/jcdd11050138), as it provides valuable insights into this issue.
Thank you for the suggestion. We found the study by Pizzino highly relevant and have incorporated it into the Discussion section (alongside other studies).
Furthermore, the abstract should be consolidated to a maximum of 250 words in accordance with the journal's guidelines.
We have made slight changes to the abstract while ensuring it stays within the 250-word limit.
Lastly, if feasible, we advise incorporating a detailed description of the various types of aortic valves implanted at baseline in the sample table, prior to their infection.
We have now included a detailed description of the various types of surgeries performed (this analysis was initially made but not inserted in the submitted manuscript). However, we would like to clarify that the main focus of this study was not to compare specific surgical techniques: the goal was to provide an overview of surgery in general, acknowledging that it encompasses a wide range of techniques, which can vary significantly in terms of aggressiveness and technical complexity.
Reviewer 2 Report
Comments and Suggestions for Authors
I would like to congratulate the authors on this work.
Please find below my comments and suggestions:
1. Line 81, please correct the word Decembre
2. Lines 84, 85 : What do mean by "connected by standardized treatment pathways" ? Could you please define?
3. Could you please clearly define the inclusion and exclusion criteria in the text?
4. Operative technique is described very shortly. What were the operative strategies in case of aortic root abscess or fistula? Did you use an Aortomitral curtain reconstruction technique with patch plastic? Please describe that important part in more detail.
5. How many patients received a stand-alone AVR, and how many a Bental? How many a combined procedure?
6. Please insert the references at the end of each sentence and not at the beginning of the next sentence.
7. How long was the mean ICU stay in this cohort? Authors only refer to this as "more or less than 2 days".
8. The Discussion should be rewritten, so that the Authors deliver better their findings. Clinical impact and take-home message? The Authors could empower their study with additional references.
3.a network comprising several
84
hospitals in the French capital connected by standardized treatment pathways.
Language editing is needed.
Author Response
I would like to congratulate the authors on this work.
Dear Reviewer,
Thank you for taking the time to read and analyze our manuscript. We greatly appreciate your thoughtful comments and positive feedback regarding our study. We also sincerely appreciate your constructive suggestions, which will undoubtedly help enhance the clarity and impact of our manuscript.
Please find below my comments and suggestions:
- Line 81, please correct the word Decembre
Corrected
- Lines 84, 85 : What do mean by "connected by standardized treatment pathways" ? Could you please define?
We understand the potential confusion. Henri Mondor is part of a network of 4 public hospitals in Paris that have cardiac surgeons on place, along with several other healthcare facilities that do not have a cardiac surgery unit. When a patient is evaluated (either at Henri Mondor or another facility) and there is suspicion of infective endocarditis, a standardized protocol is activated across all institutions. This protocol includes diagnostic tests, regardless of where the patient will eventually undergo surgery. These tests typically include transoesophageal echocardiography, chest CT with contrast, colonoscopy (to identify or exclude the “entry source” of infection), blood cultures, among others. We wanted to emphasize this point because, unlike in other settings where each hospital may have its own protocol, here we have a highly integrated network that embodies the true spirit of an Endocarditis Team, also beyond the hospital walls.
We have removed this sentence, as it could potentially cause further confusion and requires a detailed explanation (which we have already provided to you). Including it in the Discussion might distract from the main focus of the manuscript.
- Could you please clearly define the inclusion and exclusion criteria in the text?
The inclusion and exclusion criteria have been clearly defined in the revised manuscript now, specifically in the “Materials and Methods” section and in the first paragraph of “Results”.
- Operative technique is described very shortly. What were the operative strategies in case of aortic root abscess or fistula? Did you use an Aortomitral curtain reconstruction technique with patch plastic? Please describe that important part in more detail.
- How many patients received a stand-alone AVR, and how many a Bental? How many a combined procedure?
Regarding points 4 and 5, we agree with your judgment; we have now included a detailed description of the various types of surgeries performed (this analysis was initially done but not inserted in the submitted manuscript). However, we would like to clarify that the main focus of this study was not to compare specific surgical techniques: the goal was to provide an overview of surgery in general, acknowledging that it encompasses a wide range of techniques, which can vary significantly in terms of aggressiveness and technical complexity.
- Please insert the references at the end of each sentence and not at the beginning of the next sentence.
Corrected.
- How long was the mean ICU stay in this cohort? Authors only refer to this as "more or less than 2 days".
Considering the column in our database “ICU length of stay” (which is a parameter highly dependent on the hospital), we derived the following data: Mean: 3.43 days; Median: 2 days; Mode: 2 days. Therefore, we deemed it appropriate to use a cutoff of 2 days to identify patients who required a longer stay. As a result, 22 out of 54 patients (40.7%) were classified as having a prolonged ICU stay.
- The Discussion should be rewritten, so that the Authors deliver better their findings. Clinical impact and take-home message? The Authors could empower their study with additional references.
Thank you for your feedback. We have created a graphical abstract, as suggested by the Editor and your colleague-reviewer. We have also revised the Discussion section to better highlight the take-home message and have expanded it by incorporating additional references that we believe improve
Round 2
Reviewer 2 Report
Comments and Suggestions for Authors
The article is properly revised. The authors took clearly, all comments and suggestions into consideration.
Author Response
Comment 1: The article is properly revised. The authors took clearly, all comments and suggestions into consideration.
Response 1: Thank you for your positive feedback and for recognizing the revisions made to the article. We greatly appreciate your comments and suggestions during the revision phase.